# C2T: Classifier-based Token Tree Construction in Speculative Decoding

## Abstract

With the increasing scale of Large Language Models (LLMs), issues of inference latency and computational costs have become increasingly prominent. Speculative decoding methods have emerged to alleviate these challenges, but existing tree construction strategies exhibit inefficiencies in accurately preparing candidate token trees for the verification stage. To address this, we propose a plug-and-play method named **C2T** that leverages a lightweight three-feature classifier with only 241 parameters to help dynamically generate and pre-prune token trees, which is even applicable to early stopping in token sequence inference. Our approach outperforms traditional probability-based dynamic token tree construction methods while introducing negligible computational overhead. We evaluated our method on multiple benchmarks and models and showed that, when combined with SOTA methods such as EAGLE-2/3, it can reduce the number of candidate tokens by 25% without sacrificing acceptance length, resulting in a 7% to 17% speedup across models of different sizes.

## 1 Introduction

Large Language Models (LLMs) Achiam et al. (2023); Touvron et al. (2023) have shown remarkable abilities in various fields, but face significant bottlenecks in autoregressive token generation due to high memory bandwidth demands and underutilized GPU resources Patterson (2004); Shazeer (2019), as each token requires access to all model parameters Radford et al. (2019); Brown et al. (2020). To address this issue, Speculative Decoding (SD) Chen et al. (2023); Leviathan et al. (2023) has been developed, which quickly generates multiple draft tokens and verifies them all at once using the target model to maximize GPU computational capacity, and it has been applied in the latest influential LLMs Liu et al. (2024); Team (2025); Team et al. (2025).

Vanilla SD employs a chain structure for the draft tokens, and the verification process follows a topological order Chen et al. (2023); Leviathan et al. (2023). If a token is rejected, all subsequent tokens are also discarded. To overcome inefficiency, tree-structured draft tokens have been proposed Miao et al. (2024); Sun et al. (2024), which integrate multiple chains. Static tree methods, such as EAGLE-1 Li et al. (2024a) and Medusa Cai et al. (2024), use preset tree structures that bias the sampling rate of specific positions Chen et al. (2024), while dynamic methods, such as EAGLE-2 Li et al. (2024b), rely on contextual information.

For dynamic tree methods, most are designed to build and prune trees on the basis of confidence scores. The most straightforward approach is to use the joint probability as confidence Li et al. (2024b); Wang et al. (2024); Brown et al. (2024); Qin et al. (2024). However, directly using joint probability as confidence is not enough for complex situations, leading to misjudgments. We observe that using joint probability as the confidence score leads to a discrepancy with the final acceptance rate. From the perspective of tree structure, whether a token node is accepted depends not only on its own isolated attributes like probability, but also on its global properties within the entire tree. This means that incorporating more variables together is necessary in the confidence score calculation. Therefore, we propose a tree construction method based on a designed tiny classifier to perform this calculation. Our contributions are summarized below:

- We provide a statistical analysis showing that methods such as EAGLE-2, which directly use joint probability as a confidence measure, exhibit a bias with respect to the acceptance

rate. Furthermore, we rigorously prove the underlying cause of this phenomenon through mathematical derivation.

- We propose a classifier-based dynamic tree construction method that, due to its simple design and excellent transferability, can serve as a plug-and-play confidence estimator in any generation-probability-based inference process like speculative decoding.

- In combination experiments with SOTA methods EAGLE-2/3, our approach reduces the number of candidate tokens by 25% under the same acceptance length and achieves a speedup of 7% to 17% across models of different sizes.

## 2 BACKGROUND

### 2.1 SPECULATIVE DECODING

Speculative decoding (SD) Chen et al. (2023); Leviathan et al. (2023) is an algorithm designed to speed up model reasoning by leveraging the parallel computing capabilities of attention mechanisms. It consists of two main stages. The first stage is the draft phase, where a smaller model, known as the draft model $M_d$, generates draft tokens. The second stage is verification, where the draft tokens are verified all at once by the larger model, known as the target model $M_t$. At the same time, we call the draft tokens chosen to be verified as candidate tokens.

For each candidate token, let the generation probability of $M_d$ be $p$, and the verification probability of $M_t$ be $q$. Then, its acceptance probability is $\min(1, q/p)$. If the candidate token is accepted, the process proceeds to the draft and verification of the next token. Otherwise, a new token is sampled from the distribution with probability proportional to $\mathrm{norm}(\max(0, q - p))$, and the process is repeated.

### 2.2 TREE ATTENTION

Vanilla SD uses a chain structure, where if a token is rejected, all subsequent tokens are also discarded. SpecInfer's Miao et al. (2024) Tree Attention achieves the integration of multiple speculations at a minimal cost and has been widely adopted in other SD methods Sun et al. (2024); Li et al. (2024a); Cai et al. (2024); Chen et al. (2024); He et al. (2023); Svirschevski et al. (2024). In the pioneer SD works, the token tree was static, with a preset tree shape, and the draft model generated tokens layer by layer and filled in the corresponding positions. This static method is undoubtedly rough, so various heuristic dynamic generation methods appeared later Li et al. (2024b); Wang et al. (2024); Brown et al. (2024); Qin et al. (2024); Huang et al. (2024). These methods essentially employ strategies to improve the application of beam search in dynamic tree construction, and largely retain beam search's reliance on probability as the sole criterion. Some approaches introduce beam-entropy and positional information as additional signals for early stopping. Several trainable dynamic methods have also been proposed Huang et al. (2024); Mamou et al. (2024); Zhang et al. (2024); Bachmann et al. (2025). Because the classifier features they designed are too large, such as the full probability vector, the complete prompt, or the hidden state, these methods introduce non-negligible latency during the tree construction process. Therefore, they are only suitable for early stopping or pre-allocation in chain-structured inference. Overly complex features can also reduce generalizability, especially for features such as hidden states and prompts, which are highly dataset-dependent.

The current SOTA dynamic tree construction method, EAGLE-2/3 Li et al. (2024b; 2025), introduces the joint probability of each node as contextual information, and divides the sampling process into two stages: expand and rerank. The former is to build the tree, and the latter is for post-pruning. We can denote the token tree after expansion as $T_1$, and the token tree reranked as $T_2$.

## 3 MOTIVATION

### 3.1 BIAS BETWEEN PROBABILITY AND ACCEPTANCE RATE

As shown in Figure 1, we conducted entropy-based binning experiments on MT-bench using the EAGLE model, recording the generation probability, acceptance rate, and the bias between the two for the i-th highest probability sample within each entropy interval. The similarity in the distributions

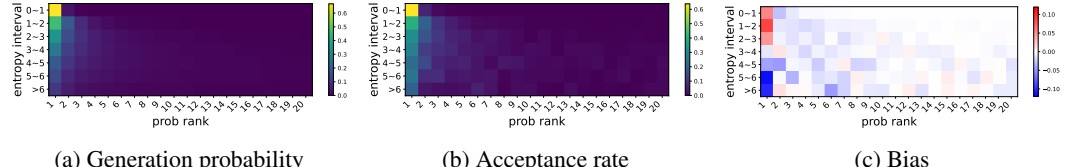

(a) Generation probability       (b) Acceptance rate       (c) Bias

Figure 1: The entropy-binned experiment of the EAGLE model on MT-bench, where the y-axis denotes different entropy intervals and the x-axis denotes the i-th highest probability sample within each entropy interval.

of Figure 1a and Figure 1b indicates a correlation between the generation probability and the final acceptance rate. However, the bias shown in Figure 1c demonstrates that solely using generation probability as the confidence score inevitably introduces a deviation.

## 3.2 THE IMPACT OF ENTROPY

As shown in Figure 1c, there are two main bias between probability and acceptance rate:

- **Weakness1**: When the entropy is low, the node with the highest probability in the distribution is overestimated in terms of probability.

- **Weakness2**: When the entropy is high, the node with the highest probability in the distribution is underestimated in terms of probability.

This can be rigorously proven as follows.

**Lemma1.** *Monotonicity of the minimal entropy with respect to $p_{\max}$.*

For $p_{\max} \in \left[\frac{1}{n}, 1\right]$, the minimal entropy is achieved when $p = \left(p_{\max}, \frac{1-p_{\max}}{n-1}, \ldots, \frac{1-p_{\max}}{n-1}\right)$, and is given by:

$$H_{\min}(p_{\max}) = -p_{\max} \log p_{\max} - (1 - p_{\max}) \log \left(\frac{1 - p_{\max}}{n - 1}\right). \quad (1)$$

Since $p_{\max} \geq \frac{1}{n}$, the derivative $\frac{dH_{\min}}{dp_{\max}} \leq 0$, which implies that $H_{\min}(p_{\max})$ is monotonically non-increasing with respect to $p_{\max}$.

**Lemma 2.** *Relationship between expected acceptance rate and total variation distance.*

Let the validation probability distribution of the target model be $q = (q_1, \ldots, q_n)$, and the generation probability distribution of the draft model be $p = (p_1, \ldots, p_n)$. The total variation distance (TVD) between $p$ and $q$ is defined as:

$$\text{TVD}(p, q) = \frac{1}{2} \sum_{i=1}^{n} |p_i - q_i| = \frac{1}{2}\left( \sum_{q_i \geq p_i} (q_i - p_i) + \sum_{q_i < p_i} (p_i - q_i) \right) = \sum_{q_i \geq p_i} (q_i - p_i) \quad (2)$$

The expected acceptance rate $E(\alpha)$ for speculative decoding is:

$$E(\alpha) = \sum_{i=1}^{n} \min\left(1, \frac{q_i}{p_i}\right) p_i = \sum_{q_i \geq p_i} p_i + \sum_{q_i < p_i} q_i = 1 - \sum_{q_i \geq p_i} (q_i - p_i) = 1 - \text{TVD}(p, q). \quad (3)$$

**Lemma3.** *Bias between $p_{\max}$ and the lower bound of expected acceptance rate.*

When $p = \left(p_{\max}, \frac{1-p_{\max}}{n-1}, \ldots, \frac{1-p_{\max}}{n-1}\right)$ and $q = (0, 1, 0, \ldots, 0)$, the total variation distance reaches its maximum:

$$\text{TVD}_{\max}(p, q) = \frac{1}{2}[p_{max} + (1 - \frac{1 - p_{max}}{n - 1}) + (n - 2)\frac{1 - p_{max}}{n - 1}] = \frac{p_{max} + 1}{2} \quad (4)$$

Therefore, the bias between $p_{\max}$ and the lower bound of the expected acceptance rate is

$$\text{Bias} = p_{\max} - E_{min}(\alpha) = p_{\max} - 1 + \text{TVD}_{\max}(p, q) = \frac{3p_{\max} - 1}{2} \tag{5}$$

**In conclusion,** combining the above lemmas, when $H$ is small, $p_{\max}$ tends to be large (**Lemma1**), resulting in a positive bias(**Lemma3**), corresponding to **Weakness1**. Conversely, when $H$ is large, $p_{\max}$ tends to be small, leading to a negative bias, corresponding to **Weakness2**.

### 3.3 THE IMPACT OF DEPTH

In addition to entropy, directly using the probability as a confidence measure also neglects the positional information of candidate tokens. Due to the topological property of speculative sampling verification—namely, if a token is rejected, all subsequent tokens will be discarded. Thus, shallower nodes should be assigned higher confidence. However, methods based solely on joint probability cannot distinguish between nodes of different depths using probability alone.

Table 1: The acceptance rates for the same joint probability intervals at different depths.

| Joint Prob | d=1 | d=2 | d=3 |
|---|---|---|---|
| **(0.0, 0.05]** | 0.032 | 0.007 | 0.006 |
| **(0.5, 0.55]** | 0.473 | 0.390 | 0.320 |
| **(0.9, 0.95]** | 0.795 | 0.701 | 0.632 |

- **Weakness3**: When nodes at different depths have similar joint probabilities, the influence of depth is effectively deactivated.

To demonstrate the importance of the depth factor, we conducted a statistical analysis. Using the same experimental configuration as in Figure 1, we collected candidate tokens with joint probabilities falling within the same interval and examined their acceptance rates at different depths. As shown in Table 1, for different nodes with joint probabilities within the same interval, nodes at shallower depths exhibit higher acceptance rates. Therefore, if we consider only probability as the confidence measure, the depth factor is neglected, which is crucial in speculative sampling.

## 4 C2T

To enable more accurate dynamic construction of token trees, we propose **C2T**: a method that utilizes a lightweight **C**lassifier **to** help construct a token **T**ree.

### 4.1 CLASSIFIER

To ensure the lightweight nature of the classifier, we designed a two-layer feed-forward network (FFN) with ReLU activation between layers. To enhance transferability, we abandoned the commonly used hidden state input in other trainable pre-pruning algorithms and instead used only three statistical features as input: joint probability, entropy, and depth. The total number of parameters in the classifier is only 241. We constructed the training set using the complete token trees (prior to recall) generated by EAGLE-2, labeling each token with its three features and whether it was ultimately accepted. Detailed training procedures are provided in Appendix C.

After training the classifier, we conducted a quantitative analysis of the three parameters, as shown in Figure 2. As shown in Figure 2a, the different surfaces indicate that tokens with higher joint probability have higher confidence scores, which is consistent with general statistical principles. However, the distances between the three surfaces exhibit a U-shape along the entropy axis: in both the high-entropy and low-entropy regions, the confidence gap between high-probability and low-probability nodes is smaller than that in the medium-entropy region. This suggests that the classifier mitigates the **Weakness1** and **Weakness2** in 3.1. As shown in Figure 2b, except for the low-entropy region where entropy is less than 1, higher entropy corresponds to higher confidence scores. This is interpretable: from a recall perspective, regions with greater uncertainty require more attention. As shown in Figure 2c, shallower nodes generally have higher confidence, which effectively addresses **Weakness3**. Moreover, this correction becomes more pronounced as the probability decreases. This is because the shallow nodes, having the same probability as the deep nodes, typically also have lower joint probabilities.

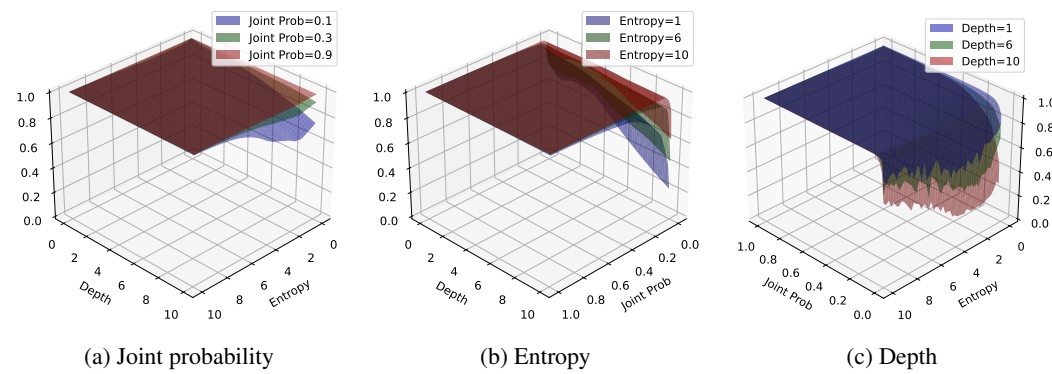

(a) Joint probability        (b) Entropy        (c) Depth

Figure 2: For the quantitative analysis of the three features, we fix one feature value and plot the interpolated surface, where the z-axis denotes the confidence score.

## 4.2 TREE CONSTRUCTION

C2T is a pre-pruning approach, which can be divided into two steps: the first pruning based on confidence, and the second pruning based on topK.

**First Pruning based on Confidence** After obtaining the classifier, we can use it to build the token tree. The tree construction based on the classifier is a layer-by-layer construction process. The draft model can obtain the generation probability of each node during each forward pass. Let a node be $i$, and the generation probability of this node be $p_i$, then the joint probability of this node is denoted as

$$P_i = \prod_{j \in Path(root, i)} p_j \tag{6}$$

Where the $Path(root, t_i)$ denotes the set of all nodes on the path from the root node to node $i$.

At this point, we can also determine the entropy of node $i$'s probability distribution.

$$H_i = -\sum_{j \in S(i)} p_j \log p_j \approx -\sum_{j \in S_{1000}(i)} p_j \log p_j \tag{7}$$

Where $S(i)$ denotes the probability distribution of $i$, and $S_{1000}(i)$ denotes the set of nodes with top 1000 probabilities. We can also easily record the depth of node $d_i$.

After obtaining these three features, we can use the trained classifier to obtain the confidence of node $i$, denoted as:

$$C_i = F(P_i, H_i, d_i) \tag{8}$$

Where $F(*)$ denotes the classifier's output. Since $C_i$ is normalized, we set a threshold $\beta$ between 0 and 1 to determine which tokens participate in generating the next tree layer. Since tuning $\beta$ involves a trade-off between acceptance length and the number of candidate tokens, this value is typically set to 0.5 unless otherwise specified. This screening-generation process repeats at each tree layer until the current depth $d_i$ reaches the maximum depth $d_{max}$ we set, or until no nodes in the current tree layer have confidence greater than $\beta$.

**Second Pruning based on Topk** To further control the size of the tree and reduce the computational overhead introduced by feature calculation and the classifier, we additionally apply top-$K$ secondary pruning at two stages:

1. To reduce the classifier's computational cost, we calculate confidence only for the Top$K$ tokens with the highest generation probability.

2. To prevent excessive tree expansion, we limit the number of tokens participating in the next tree layer's generation. After identifying the tokens that pass the classifier test in the current tree layer, we select only the Top$K$ tokens with the highest confidence as the final candidates.

The pseudocode for the overall algorithm is provided in Appendix A.

### 4.3 THEORETICAL PROOFS

**Theoretical Speed-up**  Speculative sampling proceeds in two stages: draft and verify. Let $t_{draft}$ and $t_{verify}$ be the per-token forward latencies of the draft and target models, respectively. Pre-pruning additionally spends $t_{math1}$ on input-feature extraction and $t_{cls}$ on token-level classification, while post-pruning spends $t_{math2}$ to gather contextual information.

Under the same model pair, the latency comparison leads directly to

$$\text{Speedup} = \frac{(t_{draft} + t_{math2}) + t_{verify}\alpha}{(t_{draft} + t_{math1} + t_{cls} + t_{verify})(1 - \rho)\alpha}, \quad (9)$$

where $\alpha$ is the recall ratio of candidate tokens (constant $\approx 0.1$ for EAGLE-2/3) and $\rho$ is the fraction removed by pre-pruning ($\approx 0.25$ measured).

For heuristic post-pruning $t_{math2}$ covers top-K selection and joint-probability calculation; for C2T $t_{math1}$ covers top-M selection, entropy computation, and a second top-K step (M=1000, vocabulary $V \gg M$). As shown in Appendix H, $t_{math1}, t_{math2}, t_{cls}$ are negligible versus the large-model forward passes. With the draft model being a single layer of the target model, the theoretical latency ratios are

$$t_{math1} : t_{math2} : t_{cls} : t_{draft} : t_{verify} \approx \begin{cases} 0 : 0 : 0 : 1 : 32, & 7B/8B \\ 0 : 0 : 0 : 1 : 80, & 70B \end{cases} \quad (10)$$

Inserting these into equation 9 yields theoretical speed-ups of $1.69\times$ and $1.48\times$ for 7B/8B and 70B models, respectively.

**Bias Analysis**  In practice, the draft stage is memory-bound; latency is governed by the number of forward passes (draft length). C2T mainly reduces the number of branches rather than the depth of the tree during tree-structured inference, so the required number of draft iterations remains similar to EAGLE-2/3. Hence, the measured speed-up becomes

$$\text{Speedup}' = \frac{t_{draft} + t_{math2} + t_{verify}\alpha}{t_{draft} + t_{math1} + t_{cls} + t_{verify}(1 - \rho)\alpha}. \quad (11)$$

Although $t_{cls}, t_{math1}, t_{math2}$ are FLOP-light, kernel launch and GPU utilisation make them non-negligible. Wall-clock measurements on 1×L40 (48G) for 7B/8B and 2×A100 (80G) for 70B give

$$t_{math1} : t_{math2} : t_{cls} : t_{draft} : t_{verify} \approx \begin{cases} 1 : 1 : 1 : 5 : 100, & 7B/8B \\ 1 : 1 : 1 : 10 : 400, & 70B \end{cases} \quad (12)$$

Substituting these into equation 11 produces actual speed-ups of $1.10\times$ and $1.21\times$, within 3% of our best experimental results ($1.07\times$, $1.17\times$).

## 5 EXPERIMENTS

**Models:** We used LLaMA-2-Chat 7B, 13B, 70B Touvron et al. (2023), Vicuna 7B, 13B, 33B Chiang et al. (2023), LLaMA-3.1-Instruct 8B, LLaMA-3.3-Instruct 70B Grattafiori et al. (2024) and DeepSeek-R1-Distill-LLaMA 8B Liu et al. (2024) as target model $M_t$, and the corresponding draft model $M_d$ is from EAGLE-2/3 Li et al. (2024a; 2025) and LLaMA-3.2-Instruct 1B Grattafiori et al. (2024).

**Tasks:** To compare with EAGLE-2/3 Li et al. (2024b), we aligned with it on the dataset. For tasks such as multi-round dialogue, code generation, mathematical reasoning, instruction following,

summarization, and Q&A, we selected the MT-bench Zheng et al. (2023), HumanEval Chen et al. (2021), GSM8K Cobbe et al. (2021), Alpaca Taori et al. (2023), CNN/Daily Mail Nallapati et al. (2016), and Natural Questions Kwiatkowski et al. (2019), respectively.

**Metrics:** We mainly focus on the following indicators:

- **Decoding speed** $v$: The total number of new tokens divided by wall-clock time.
- **The number of candidate tokens** $\gamma$: The total number of tokens verified by $M_t$.
- **Accept length** $\tau$: The average length accepted by $M_t$ for each generation.

**Comparison:** We evaluate the improvements of our method when combined with vanilla speculative decoding Chen et al. (2023); Leviathan et al. (2023) as well as with the widely recognized industry baselines EAGLE-2 Li et al. (2024b) and EAGLE-3 Li et al. (2025) (EAGLE-3 still adopts the dynamic tree construction approach from EAGLE-2).

**Environment:** All experiments for 7B/8B and 13B models were conducted on $1 \times$ L40 (48G) GPU, while all experiments for 70B models were conducted on $2 \times$ A100 (80G) GPUs.

## 5.1 SPEEDUP EFFECTIVENESS

To demonstrate the plug-and-play performance improvement of C2T, we directly compare its gains when combined with Vanilla SD and EAGLE-2/3, without additional parameter tuning. Vanilla SD uses LLaMA-3.2-Instruct 1B as the draft model, and EAGLE-2/3 follow their original settings. When combined with C2T, we increase the draft length/depth by two to better utilize its tree construction capability. Results show that C2T enables deeper token trees with fewer candidate tokens. All evaluations are conducted at temperature = 0, and due to the lack of an EAGLE-3 model for LLaMA-2, we report results only on LLaMA-3. The threshold parameter $\beta$ of C2T is set to 0.5.

Table 2: Performance metrics of different models on multiple benchmarks. L31 8B denotes LLaMA-3.1-Instruct 8B, DSL 8B denotes DeepSeek-R1-Distill-LLaMA 8B, L33 70B denotes LLaMA-3.3-Instruct 70B, SD denotes vanilla speculative decoding using LLaMA-3.2-Instruct 1B as draft model, E2 denotes EAGLE-2, E3 denotes EAGLE-3, +C denotes combined with C2T, $\gamma$ denotes the candidate tokens and is measured in units of one thousand tokens, $\tau$ denotes accept length, $v$ denotes the decoding speed and is measured in units of tokens per second.

| Model | Method | MT | | | Humaneval | | | GSM8K | | | Alpaca | | | CNN/DM | | | QA | | | Avg | |
|---|---|---|---|---|---|---|---|---|---|---|---|---|---|---|---|---|---|---|---|---|---|
| | | $\gamma$ | $\tau$ | $v$ | $\gamma$ | $\tau$ | $v$ | $\gamma$ | $\tau$ | $v$ | $\gamma$ | $\tau$ | $v$ | $\gamma$ | $\tau$ | $v$ | $\gamma$ | $\tau$ | $v$ | $v$ | $\Delta$ |
| | SD | **88** | 3.69 | 39 | **39** | 4.21 | 50 | **24** | 3.71 | 38 | 27 | 3.78 | 39 | **37** | 3.20 | 28 | **25** | 3.19 | 28 | 37 | |
| | SD+C | 90 | **4.06** | **44** | 41 | **4.86** | **58** | 26 | **4.13** | **42** | **27** | **4.26** | **43** | 38 | **3.43** | **29** | 27 | **3.45** | **30** | **41** | ↑ 11% |
| L31 8B | E2 | 788 | 4.04 | 59 | 357 | 4.69 | 76 | 218 | 4.25 | 70 | 251 | 4.13 | 78 | 376 | 3.45 | 55 | 226 | 3.60 | 60 | 66 | |
| | E2+C | **711** | **4.08** | **65** | **315** | **4.78** | **81** | **201** | **4.25** | **74** | **230** | **4.21** | **82** | **323** | **3.52** | **61** | **185** | **3.65** | **66** | **71** | ↑ 8% |
| | E3 | 522 | 6.19 | 81 | 247 | 6.74 | 109 | 148 | 6.23 | 104 | **154** | 6.72 | 98 | 228 | 5.38 | 87 | 154 | 5.23 | 88 | 94 | |
| | E3+C | **461** | **6.34** | **86** | **189** | **7.06** | **120** | **143** | **6.46** | **110** | 155 | **7.01** | **107** | **192** | **5.46** | **94** | **130** | **5.28** | **93** | **101** | ↑ 7% |
| | SD | **146** | 3.27 | 31 | **67** | 3.66 | 40 | **49** | 4.09 | 42 | 73 | 3.27 | 31 | 78 | 2.95 | 25 | 75 | 2.88 | 24 | 32 | |
| | SD+C | 148 | **3.56** | **34** | 69 | **4.05** | **45** | 54 | **4.68** | **48** | **69** | **3.56** | **35** | **72** | **3.13** | **27** | **69** | **3.03** | **26** | **36** | ↑ 12% |
| DSL 8B | E2 | 788 | 3.81 | 61 | 467 | 4.29 | 80 | 431 | 4.4 | 81 | 669 | 3.77 | 66 | 590 | 3.33 | 60 | 558 | 3.77 | 66 | 69 | |
| | E2+C | **720** | **3.92** | **68** | **401** | **4.55** | **87** | **388** | **4.73** | **87** | **609** | **3.92** | **71** | **541** | **3.49** | **64** | **520** | **3.81** | **71** | **74** | ↑ 7% |
| | E3 | 838 | 5.73 | 95 | 377 | 6.55 | 110 | 289 | 7.04 | 118 | 411 | 5.59 | 94 | 462 | 5.01 | 82 | 439 | 4.99 | 85 | 97 | |
| | E3+C | **805** | **5.93** | **101** | **359** | **6.87** | **121** | **276** | **7.49** | **130** | **391** | **5.66** | **98** | **459** | **5.07** | **87** | **404** | **5.04** | **89** | **104** | ↑ 7% |
| | SD | 96 | 3.72 | 12 | **43** | 4.26 | 18 | 22 | 3.82 | 14 | **29** | 3.72 | 12 | 38 | 3.15 | 12 | 32 | 2.87 | 7 | 12 | |
| | SD+C | **92** | **4.11** | **14** | 44 | **4.87** | **23** | **20** | **4.26** | **18** | 31 | **4.11** | **15** | **35** | **3.43** | **15** | **25** | **3.15** | **10** | **16** | ↑ 33% |
| L33 70B | E2 | 275 | 3.19 | 22 | 335 | 4.44 | 30 | 200 | 3.53 | 24 | 275 | 3.19 | 22 | 290 | 3.51 | 22 | 275 | 2.77 | 19 | 23 | |
| | E2+C | **262** | **3.29** | **25** | **321** | **4.59** | **34** | **185** | **3.62** | **31** | **262** | **3.29** | **25** | **288** | **3.95** | **27** | **271** | **2.83** | **22** | **27** | ↑ 17% |
| | E3 | 138 | 6.25 | 43 | 222 | 6.59 | 45 | 113 | 6.14 | 42 | 138 | 6.25 | 43 | 203 | 5.01 | 30 | 160 | 4.75 | 33 | 39 | |
| | E3+C | **114** | **6.64** | **47** | **190** | **6.91** | **52** | **84** | **6.39** | **52** | **114** | **6.64** | **47** | **217** | **5.12** | **33** | **148** | **4.84** | **37** | **45** | ↑ 15% |

As shown in Table 2, under fixed parameter settings, for chain-structured inference with vanilla SD, C2T can maintain an almost constant $\gamma$ even with longer draft lengths, thereby achieving acceleration.

For tree-structured inference with EAGLE-2/3, integrating C2T enables simultaneous optimization of both $\tau$ and $\gamma$, with the performance gains becoming more pronounced as the size of the target model increases.

## 5.2 TRANSFERABILITY

Since the C2T classifier is trained entirely on statistical features, it exhibits strong transferability across both datasets and models. We first train a classifier using LLaMA-2-Chat 7B on MT-bench, then freeze its parameters and directly apply it to other datasets and models for C2T experiments. The results, compared with EAGLE-2 and C2T using a retrained classifier, are shown in Figure 3.

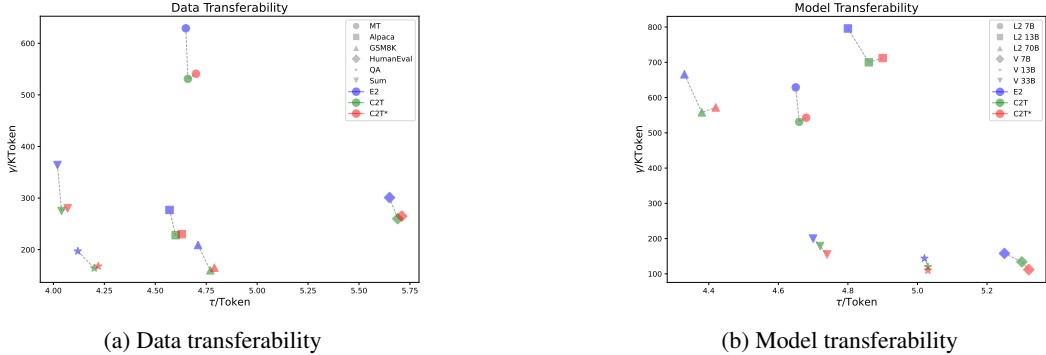

| (a) Data transferability | (b) Model transferability |

Figure 3: For the transferability scatter plot, L2 indicates LLaMA-2 and V indicates Vicuna. E2 indicates EAGLE-2, C2T* indicates C2T using a re-trained classifier. C2T and C2T* consistently contribute a higher accept length and lower candidate tokens over EAGLE-2.

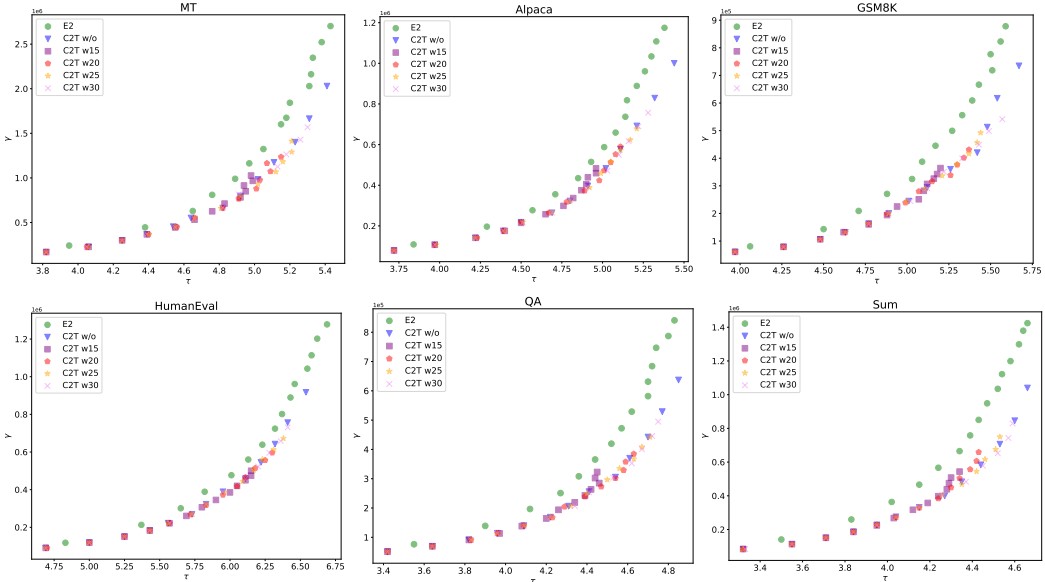

Figure 4: The scatter plot uses the LLaMA-2-Chat 7B model, with the acceptance length $\tau$ on the x-axis and the number of candidate tokens $\gamma$ on the y-axis. E2 represents EAGLE-2, w/o represents not using topK secondary pruning, and w$K$ represents the use of Top$K$ secondary pruning with $K$ values of 15, 20, 25, and 30.

It can be observed that C2T demonstrates excellent transferability across different datasets, as well as strong transferability across models of different sizes within the same family. Compared to C2T with a retrained classifier, the transferred classifier yields a slightly lower acceptance length ($\tau$), but also results in a lower number of candidate tokens ($\gamma$), leading to comparable overall performance.

When directly transferring to a different model family (e.g., LLaMA-2 to Vicuna), the performance decreases but still remains superior to EAGLE-2. Retraining the classifier on the new model can further enhance the advantage.

## 5.3 VARIABLE EXPERIMENTS ON $\beta$ AND TOP-$K$.

During C2T-based pruning, two key parameters are involved: the threshold $\beta$ for the classifier to filter tokens, and the top-$K$ parameter for the secondary tree constraint. We conduct experiments using the LLaMA-2-Chat 7B model on different datasets. For fair comparison with EAGLE-2, we align the maximum tree depth with $d_{max} = 10$. We then vary EAGLE-2's final recall parameter top-$N$, as well as C2T's $\beta$ and top-$K$, to produce the following Figure 4. It can be observed that, regardless of how $\beta$ and top-$K$ are adjusted, C2T consistently outperforms EAGLE-2.

## 5.4 C2T IN SGLANG

Table 3: The acceleration ratios of C2T under different batch sizes on SGLang are summarized below. Inference is performed on a mixed dataset of 240 samples, consisting of 80 samples from each benchmark: Alpaca, CNN/Daily Mail, and GSM8k. The results include two metrics: total inference time (seconds) and relative acceleration ratio.

| Env. | Method | BS=64 | | BS=128 | | BS=256 | |
|---|---|---|---|---|---|---|---|
| | Baseline | 38.05 | 1.00X | 34.90 | 1.00X | 32.17 | 1.00X |
| L31-8B / 1 × L40 | EAGLE | 33.41 | 1.13X | 34.47 | 1.01X | 32.82 | 0.98X |
| | C2T | **31.79** | **1.19X** | **31.12** | **1.12X** | **29.24** | **1.10X** |
| | Baseline | 126.22 | 1.00X | 99.93 | 1.00X | 98.84 | 1.00X |
| L3-70B / 4 × A100 | EAGLE | 90.37 | 1.40X | 81.24 | 1.23X | 80.77 | 1.22X |
| | C2T | **84.44** | **1.49X** | **71.68** | **1.39X** | **71.82** | **1.37X** |

We evaluate the performance of C2T on SGLang with a batch size greater than 1 (bs > 1), using sequential speculative decoding. For comparison, we use EAGLE with a static draft length as the baseline. In this setting, C2T primarily contributes to early stopping. The benchmark is a mixture of GSM8K, Alpaca, and CNN/DailyMail datasets, with the maximum draft length set to 4. The speedup is defined as the ratio of the total runtime of the current method to that of standard autoregressive generation. The results are shown in Table 3. C2T performs slightly worse at smaller batch sizes, as the number of candidate tokens in sequential speculative decoding is limited and does not reach the computation-bound of the target model verification. However, as the batch size increases, the benefits of pruning with C2T become more pronounced.

## 6 CONCLUSION

In this paper, we propose a plug-and-play optimization method, C2T, to replace the use of joint probability alone as the confidence measure, thereby improving dynamic tree construction in speculative sampling. We analyze the limitations of heuristic methods that rely solely on joint probability for confidence estimation, and demonstrate the superiority of C2T through both theoretical analysis and empirical results. As a plugin, C2T can be integrated with EAGLE-2/3 for pruning, or combined with vanilla speculative decoding to enable early stopping in chain-structured inference, and requires training only a lightweight classifier with 241 parameters, yet is able to reduce the number of candidate tokens for final verification by 25% while maintaining the same acceptance length, resulting in a 7% to 17% speedup.

**Limitation** Compared to methods using joint probability directly, C2T adds computation overhead. However, this overhead is negligible when verification time is significant. The quantitative analysis shows that the additional FLOPs introduced by C2T are negligible and imply potential for further optimization in engineering implementation. Please refer to Appendix H for analysis details.

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

# A  C2T ALGORITHM

---

**Algorithm 1:** C2T

---

**Input:** draft model $M_d$, root node $r$, maximum depth $d_{max}$, threshold $\beta$
**Result:** token tree $T$
current depth $d \leftarrow 0$
confidence set $C \leftarrow \{C_r = 1\}$
token tree $T \leftarrow \{r\}$
current tree layer node set $N \leftarrow \{r\}$
**while** $d < d_{max}$ *and exists* $C_i > \beta$ *in* $C$ **do**
    $N \leftarrow topK(M_d(N))$
    $C \leftarrow \{\}$
    **for** $i$ *in* $N$ **do**
        $P_i \leftarrow \prod_{j \in Path(root,i)} p_j$
        $d_i \leftarrow d$
        $H_i \leftarrow -\sum_{j \in S_{1000}(i-1)} p_j \log p_j$
        $C_i \leftarrow F(P_i, H_i, d_i)$
        Append $C_i$ to $C$
        **if** $C_i < \beta$ **then**
            Remove node $i$ from $N$
        **end**
    **end**
    $d \leftarrow d + 1$
    extend $N$ or $topK(N)$ to $T$
**end**
**return** $T$

---

# B  EAGLE-1 AND EAGLE-2

We conducted our experiments on LLaMA-2 7B and MT-bench. According to Table 1 of EAGLE-2 Li et al. (2024b), the accept length of EAGLE-1 Li et al. (2024a) is 3.62, and that of EAGLE-2 is 4.70. We also obtained similar results in our reproduction in Table 4. However, during the experiment, we found that this comparison is not fair. Based on Appendix A of the EAGLE-1 paper and the EAGLE-2 paper, we can obtain the shapes of their respective token trees, where the size of the token tree in EAGLE-1 is 25, while that in EAGLE-2 is 60. Therefore, we further aligned the sizes of the token trees of both models and introduced our method C2T. The experimental results are shown in Table 4.

After aligning the tree sizes, EAGLE-2's performance falls short of expectations, with only an 8% improvement in accept length. In practice, dynamic methods incur additional costs, leading to worse wall-clock times. While dynamic methods generate more accurate token trees than static methods, the extra computational cost means the tree size must be increased to achieve a speedup. Essentially, dynamic methods optimize GPU utilization further, as manually designing larger trees is extremely difficult and impractical for complex scenarios. However, given GPU limitations, increasing tree size also increases the verification burden on the target model. Thus, C2T's ability to generate more compact trees while maintaining the same accept length is particularly valuable.

Table 4: For EAGLE-2 and C2T, Top$K$=10 and $d_{max}$=6.

| method | setting | $\tau$ | $\gamma$ |
|--------|---------|--------|----------|
| E-1 | Top$N$=25 | 3.66 | 340054 |
| E-2 | Top$N$=25 | 3.95 | 317668 |
|     | Top$N$=60 | 4.65 | 628980 |
| C2T | $\beta$=0.85 | 4.06 | 226569 |
|     | $\beta$=0.65 | 4.66 | 531947 |

## C    CLASSIFIER TRAINING

### C.1    DATASET

The classifier used in this paper, if not specifically mentioned, is trained on the token tree generated by LLaMA-2 7B on the MT-bench using the EAGLE-2 strategy. The settings are $d_{max} = 10$, Top$K = 10$, and the Top$N = 1011$ (meaning no recall is performed during the rerank stage, and the complete tree is used as the training dataset). Each data entry uses joint probability, depth, and entropy as features, and whether it is accepted as the label. We simply clean the data by dropping entries containing NA values, resulting in 8880 token trees, each containing 1011 nodes, for a total of 8880 * 1011 = 8,977,680 training data entries.

### C.2    TRAINING AND EVALUATION

We split the dataset in a ratio of 0.95:0.05. Since the dataset has sparse positive samples, for example, each token tree has 1011 nodes, however, only 3.5 tokens are accepted by the $M_t$, so we perform negative sampling on this sparse dataset. During training, we set the batch size to 1024, and during evaluation, to align with the token tree verification configuration, we set the batch size to 1011. We use Adam as the optimizer with a learning rate (lr) of $1 \times 10^{-3}$ and train for 10 epochs, and use BCE as the criterion.

For evaluation, we focus more on recall, meaning the classifier should try to recall all tokens that are ultimately accepted by $M_t$. At the same time, we should also pay attention to the positive rate, which is the probability that the classifier predicts a token as positive, denoted as $\theta$. This value corresponds to the ratio of Top$N$ to the size of $T_1$ in EAGLE-2 and is positively correlated with the final $\gamma$. When selecting the classifier, priority should be given to models with significantly higher recall. Among models with similar recall, choose the one with a smaller $\theta$.

### C.3    STRUCTURE

In this paper, we also briefly explored the effects of classifiers with different FFN structures. We mainly discussed the performance of FFNs with two layers and different hidden states, setting the hidden state of the classifier to 2, 6, 12, 24, 36, and 48, respectively. We trained various FFNs according to the training configuration above, and the training process is shown in Figure 5:

Furthermore, we applied these classifiers to the C2T inference of LLaMA-2 7B on MT-bench, aligning the threshold $\beta = 0.5$. It can be observed that, with similar $\gamma$, the $\tau$ for the six classifier structures are 3.91, 3.97, 4.02, 4.03, 4.02, and 4.02, respectively. Therefore, it can be concluded that for our task, a hidden state of 12 to 48 is more appropriate for the classifier. In our other experiments, this value is set to 48 by default.

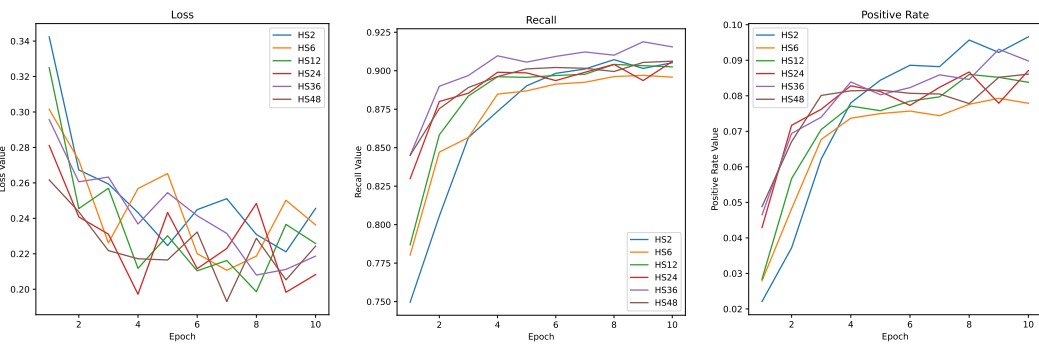

Figure 5

## D ENTROPY BUCKET VALIDATION

It can be seen that C2T effectively addresses the differentiation problem across different entropy intervals. Therefore, on the validation set, we group nodes into buckets according to their entropy values. We simulate EAGLE-2 by directly recalling the top 60 tokens, and simulate the pre-pruning process of C2T by first filtering with the classifier and then performing a second filtering step that excludes child nodes whose parent nodes have already been filtered out. For each bucket, we calculate the accuracy, precision, recall, and F1 score, and indicate the proportion of nodes in each bucket.

Table 5: Performance Comparison for Different Entropy Ranges

| Entropy Range | Proportion | Method | Accuracy | Precision | Recall | F1 Score |
|---|---|---|---|---|---|---|
| [0, 1) | 0.4293 | E2 | 0.9127 | 0.1367 | 0.9490 | 0.2390 |
| | | C2T | 0.9466 | 0.2067 | 0.9492 | 0.3395 |
| [1, 2) | 0.2166 | E2 | 0.8657 | 0.0386 | 0.8435 | 0.0738 |
| | | C2T | 0.9382 | 0.0810 | 0.8438 | 0.1477 |
| [2, 3) | 0.1570 | E2 | 0.8604 | 0.0277 | 0.8288 | 0.0538 |
| | | C2T | 0.9456 | 0.0689 | 0.8292 | 0.1273 |
| [3, 4) | 0.0954 | E2 | 0.8742 | 0.0219 | 0.8172 | 0.0427 |
| | | C2T | 0.9552 | 0.0598 | 0.8172 | 0.1114 |
| [4, 5) | 0.0515 | E2 | 0.8916 | 0.0210 | 0.8209 | 0.0409 |
| | | C2T | 0.9636 | 0.0604 | 0.8209 | 0.1125 |
| [5, 6) | 0.0256 | E2 | 0.9097 | 0.0203 | 0.7937 | 0.0396 |
| | | C2T | 0.9713 | 0.0619 | 0.7937 | 0.1149 |
| [6, ∞) | 0.0242 | E2 | 0.9329 | 0.0233 | 1.0000 | 0.0456 |
| | | C2T | 0.9802 | 0.0764 | 0.7845 | 0.1392 |
| Total | 1.0000 | E2 | 0.8900 | 0.0693 | 0.9137 | 0.1289 |
| | | C2T | 0.9478 | 0.1367 | 0.9140 | 0.2378 |

As shown in Figure 5, C2T can significantly increase precision while maintaining recall, especially in high-entropy intervals.

## E CROSS VALIDATION

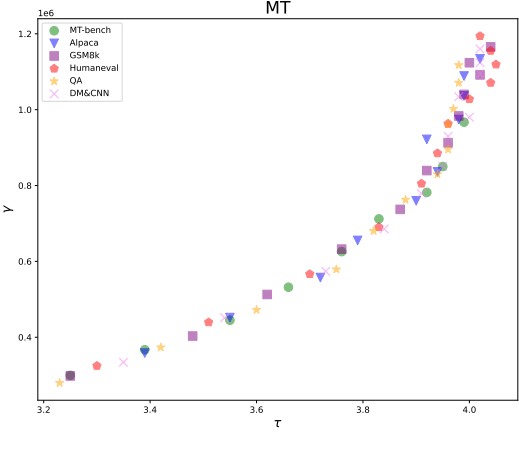

Figure 6

Since our previous experiments involved inferring token trees on MT-bench to train the classifier, and then transferring the classifier to other datasets for speculative decoding, we now cross-validate

the effectiveness of C2T on MT-bench. To do this, we train the classifier from scratch using token trees generated from other datasets and then apply it to inference on MT-bench. This experiment was conducted on LLaMA-2 7B. The results are shown in Figure 6. Classifiers trained on other datasets and those trained directly on MT-bench show nearly identical distributions in the scatter plots when used for inference with C2T. This cross-validates the feasibility of C2T on the MT-bench.

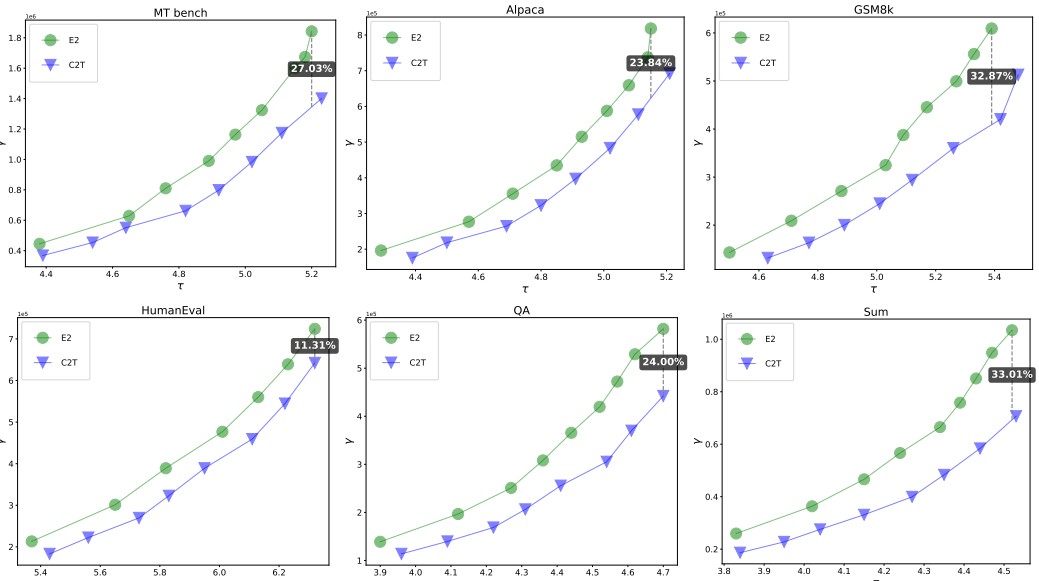

Figure 7: The line chart compares EAGLE-2 and C2T on LLaMA-2-Chat 7B, where the x-axis ($\tau$) represents the acceptance length and the y-axis ($\gamma$) denotes the number of candidate tokens. By varying the top-$N$ parameter in EAGLE-2 and the $\beta$ parameter in C2T, we plot the curves.

## F  PRUNING EFFECTIVENESS

As demonstrated by the theoretical analysis in Section 4.3, the practical benefits of C2T primarily stem from its reduction in the number of candidate tokens $\gamma$. In Experiment 5.1, to establish the reliability of C2T, we fixed the experimental parameters and did not deliberately align the acceptance length. In this section, to showcase the pruning effect of C2T on candidate tokens under the same accept length $\tau$, we control for top-$K$ and $d_{max}$ in both C2T and EAGLE-2 to ensure that pruning is performed on trees of the same size. We then vary the threshold parameter $\beta$ in C2T and the recall parameter top-$N$ in EAGLE-2 to plot the acceptance length vs. candidate token curves, as shown in Figure 7. For the same $\tau$, C2T reduces $\gamma$ by an average of 25% in different datasets.

## G  PROOF OF SIMPLIFIED CALCULATION

The method proposed in this paper, when calculating entropy, requires first obtaining the top$M$ probabilities and then calculating the entropy of these $M$ probabilities. Let the vocabulary size be $V$, and it is known that the implementation of torch.topk is based on the quickselect algorithm.

If we consider only the calculation of entropy, obtaining the top$M$ probabilities first and then calculating the entropy is more complex than directly calculating the entropy.

***Proof-1***. The FLOPs for directly calculating the entropy is $F_1 = 2 * V$. In contrast, obtaining the top$M$ probabilities involves $F_2 = C_1 * V$, after calculating the entropy of $M$ probabilities involves $F_3 = F_2 + 2 * M = C_1 * V + 2 * M$, where $C_1 > 2$ in most cases. So $2 * V < C_1 * V + 2 * M$, which means $F_1 < F_3$. In summary, selecting first and then calculating the entropy is more complex than directly calculating the entropy in most cases. □

However, we need to take into account the impact of the joint probability calculation step. Both C2T and EAGLE-2 only need to calculate the joint probabilities of the Top$K$ when computing joint probabilities. We first argue the necessity of this step.

***Proof-2***. If we were to fully calculate the joint probabilities and then select, since each tree layer has at most $K$ nodes, considering the parallel computing capability of GPUs, the overall complexity for calculating the joint probabilities is $O(V)$. There would be $K*V$ probabilities in total, and selecting the Top$K$ from them would involve $O(K*V)$. Therefore, the total complexity would be $O(V + K*V)$. In contrast, by only taking the Top$K$ for each node, due to the parallel computing nature of GPUs, the total complexity for selection is $O(V)$, resulting in $K^2$ probabilities. The complexity for selecting the Top$K$ from these probabilities is $O(K^2)$, so the overall complexity is $O(V + K^2)$. Since $K^2 << V$, therefore $O(V + K*V) > O(V + K^2)$. In summary, it is necessary to first select and then calculate the joint probabilities. $\square$

Therefore, what we are actually comparing are the complexities of the following two scenarios:

- Directly calculating the entropy and then selecting the Top$K$ probabilities.
- First selecting the top$M$ probabilities, then calculating the entropy of these $M$ probabilities, and finally selecting the Top$K$ probabilities from these $M$ probabilities.

***Proof-3***. From ***Proof-1***, we know that for the first scenario, the FLOPs before taking the Top$K$ is $F_1 = 2*V$, and the FLOPs for taking the Top$K$ is $F_4 = C_2 * V$. Therefore, the total FLOPs is $F_5 = (2 + C_2) * V$. For the second scenario, the FLOPs before taking the Top$K$ is $F_3 = C_1 * V + 2 * M$, and the FLOPs for taking the Top$K$ from the $M$ probabilities is $F_5 = C_3 * M$. Therefore, the total FLOPs is $F_6 = C_1 * V + (2 + C_3) * M$. Since for the quickselect algorithm, the final number of computations is independent of the number of elements to be selected, $C_1 \approx C_2$. Also, since $V >> M$, we have $F_4 - F_6 = (C_2 - C_1) * V + 2 * V - (2 + C_3) * M \approx 2 * V - (2 + C_3) * M > 0$, which means $F_4 > F_6$. In summary, considering the computation of joint probabilities, the FLOPs of the first scenario are more than the second scenario. $\square$

We have demonstrated the necessity of simplifying the calculation of entropy, and ***Proof-3*** implies that, from the perspective of reducing FLOPs, $M$ should be as small as possible. However, an excessively small $M$ may lead to the long-tail effect. Therefore, we conducted experiments on $M$ using LLaMA-2 7B on the MT-bench with $\beta = 0.5, topK = 15$, and the results are shown in Table 6. The results indicate that when $M = 1000$, the impact of the long-tail effect is almost completely eliminated.

## H  ADDITIONAL OVERHEAD

From the perspective of quantitative analysis, the additional computational overhead introduced by C2T compared to EAGLE-2 primarily consists of the calculation of entropy, the computation of depth, and the output of the classifier. Since the computation of joint probability is inherently coupled with the EAGLE-2's generation process, and depth can be directly obtained during the construction of the tree structure, both can be considered negligible. Therefore, the focus of our analysis is on the computational complexity of entropy calculation and the forward pass of the classifier. In addition, when calculating the entropy, we first select the top$M$ values before computing the entropy. Similarly, for EAGLE-2, the engineering implementation of calculating joint probability also requires selecting the Top$K$ values first. According to Appendix G ***Proof-3***, the costs of these two parts can offset each other. Therefore, we only need to consider the complexity introduced by calculating the entropy over $M$ values, which is $O(M)$.

Then, considering the worst-case scenario, where $K$ tokens participate in tree construction at each tree layer with a maximum

Table 6: The experiments on the values of $M$ using C2T with LLaMA-2 7B on MT-bench, with $\beta = 0.5, topK = 15$, where / indicates no use of top$M$ for simplified calculation.

| $M$ | $\tau$ | $\gamma$ |
|---|---|---|
| 100 | 3.70 | 815365 |
| 500 | 3.89 | 796981 |
| 1000 | 3.92 | 781613 |
| 10000 | 3.92 | 782104 |
| / | 3.92 | 781806 |

depth of $d_{max}$, the total complexity for entropy calculation is $O(K * d_{max} * M)$.

Given that our classifier consists of two layers with a hidden layer size of $h$:

- The complexity for matrix multiplication from the input to the hidden layer is $O(3h)$.

- The activation function computation in the hidden layer is $O(h)$.

- The matrix multiplication from the hidden layer to the output layer is $O(h)$

Under the worst-case scenario, the total classification complexity is $O(K * d_{max} * 5h)$.

In summary, the overall additional complexity introduced by C2T is $O(K * d_{max} * (M + 5h))$.

In practical scenarios, with $M = 1000$, $K = 15$, $d_{max} = 10$, $h = 48$, the complexity for entropy calculation is $O(150,000)$, and the complexity for the classifier is $O(36,000)$. The total additional overhead complexity is $O(186,000)$.

This complexity is almost negligible compared to the complexity of a single forward pass of LLMs. Compared with our experimental results, it implies a great space for engineering optimization.

## I  BENEFITS IN CHAIN MODE OF EAGLE

Experiments in chain mode are meaningful because the current dynamic tree construction does not support batch sizes greater than 1. This is due to the inability to have different attention masks within the same batch. In contrast, a chained token tree is always compatible with multi-batch scenarios.

In chain mode, C2T degrades as an early exit strategy, and EAGLE-2 is rendered ineffective, essentially reverting to EAGLE-1. We varied the maximum draft length from 5 to 9 tokens for EAGLE-1/2. Additionally, we compared dynamic methods using the maximum probability and joint probability as early stopping criteria. The results, shown in Table 7, demonstrate that C2T retains an advantage in chain mode.

Table 7: The comparison between other methods and C2T with $\beta = 0.85$ in chain mode using the LLaMA-2 7B model on the MT-bench. DyMax and DyJoint represent the dynamic methods using the maximum probability with threshold=0.3 and joint probability with threshold=0.08 as the criterion for early stopping, respectively. The maximum depth for all the dynamic methods is 10. Avg length denotes the average generation length without the initial token.

| Method | Avg length | $\tau$ | $\gamma$ |
|---|---|---|---|
| | 5 | 1.98 | 112231 |
| | 6 | 2.06 | 125992 |
| EAGLE 1/2 | 7 | 2.10 | 140238 |
| | 8 | 2.09 | 155420 |
| | 9 | 2.13 | 184140 |
| DyMax | 6.10 | 2.06 | 127018 |
| DyJoint | 6.33 | 2.12 | 129876 |
| C2T | 5.46 | 2.12 | 116694 |

## J  THE USE OF LARGE LANGUAGE MODELS

In the preparation of this paper, we utilized large language models (LLMs) to assist in the following aspects:

- **Language Polishing:** We used LLMs (e.g., GPT-4) to improve the clarity, grammar, and fluency of certain sections of the manuscript. The scientific content, technical contributions, and experimental results were entirely conceived, designed, and verified by the authors.

- **LaTeX Formatting Assistance:** LLMs were used to help generate LaTeX code for tables, figures, and appendix formatting. All content, structure, and data were provided and verified by the authors.

We emphasize that **no novel scientific content or claims were generated by LLMs**. All experiments, analyses, and conclusions were independently conducted and verified by the authors. The use of LLMs was limited to auxiliary tasks to improve presentation efficiency and clarity.

This statement is included in accordance with the transparency guidelines of the conference/journal, and to acknowledge the auxiliary role of LLMs in the writing process.

