# OpenReview forum: "C2T: Classifier-based Token Tree Construction in Speculative Decoding"
_ICLR.cc/2026/Conference — ICLR 2026 Conference Withdrawn Submission_

### Official Review · Reviewer_s15Z · 2025-10-16

**Soundness:** 3
**Presentation:** 3
**Contribution:** 2
**Rating:** 4
**Confidence:** 5

**Summary:**

The authors propose a plug-and-play method named C2T that leverages a lightweight three-feature classifier with only 241 parameters to help dynamically generate and pre-prune token trees, which is even applicable to early stopping in token sequence inference. The paper outperforms traditional probability-based dynamic token tree construction methods while introducing negligible computational overhead. Experimental results show the effectiveness of the proposed method.

**Strengths:**

This paper is technically sound and easy to understand.

The experimental results show the effectiveness of the proposed method.

**Weaknesses:**

The paper focuses on generating tree structure and improve the overall MAT and speedup the large language model.

However, they do not report the framework they use. Here comes a problem that the method may not have such speedup on the popular inference framework such as vLLM. In fact, HuggingFace Transformers framework does not optimize the speed of LLMs very well, which makes the ratio of the latency of tree generation process smaller. When using vLLM framework where the operations in LLMs are optimized very well, the tree generation process will take more time and reduce the speedup.

The author should verify their method on such inference frameworks to show that their method is actually useful in reality.

**Questions:**

See weaknesses above.

---

> ### Author Response · Authors · 2025-11-14
>
> Dear Reviewer s15Z,
>
> Although we have decided to withdraw our manuscript, I am still very willing to address some of your concerns based on the experimental results we already have. Before the formal withdrawal, we also welcome any further questions you may have.
>
> W1.The author should verify their method on such inference frameworks to show that their method is actually useful in reality.
>
> Although we did not conduct experiments on vLLM, the experiments in Section 5.4 using SGLang should also demonstrate the effectiveness of C2T within an inference framework. In fact, although the primary optimization target of C2T is pruning on a tree structure, it can also be adapted for chain mode speculative decoding within inference frameworks, as we did in Section 5.4. For example, in a scenario with a batch size of 256, if we draft 4 times per batch, we obtain 1024 tokens, which can likewise be pruned using C2T.

---

### Official Review · Reviewer_surw · 2025-10-17

**Soundness:** 1
**Presentation:** 1
**Contribution:** 1
**Rating:** 0
**Confidence:** 5

**Summary:**

This paper presents C2T, a method for draft tree construction in speculative decoding. C2T trains a very small classifier to predict the confidence of a node in the draft token tree, and uses this prediction to prune the draft tree.

Experiments with three backbone models on multiple datasets show that C2T brings further speed up over speculative decoding, EAGLE-2, and EAGLE-3.

**Strengths:**

- It's interesting that the classifier, trained only on LLaMA-2 statistics, can be applied to other models, specifically Vicuna. However, the reason behind this is not sufficiently explored or discussed - Vicuna are trained from LLaMA-2 to begin with. So Section 5.2 should analyze models from other families, e.g. Qwen or DeepSeek, for better insights.

- It's good to know that the proposed method works on SGLang.

**Weaknesses:**

Overall, the paper has poor quality that's far below the standards of ICLR.

- The paper is unprofessionally written. References are not cited in proper format throughout the paper, when it has been clearly instructed in the ICLR template.

- The paper is full of technical inaccuracies, factual errors, and overclaims.

  - The first two sentences in the paper alone (Line 11-13) already lead to overclaims - "issues of inference latency and computational costs ... Speculative decoding methods have emerged to alleviate these challenges". In fact, speculative decoding only alleviates the issue of latency at increased computational costs. In speculative decoding, LLMs verify tokens parallelly instead of autoregressively. The computational cost does not change here (if there's only one draft token at each depth of the draft tree). The increase comes from 1) the computation of draft models, and 2) the verification of additional draft tokens in tree-based draft methods.

  - The paper states "Speculative decoding is designed to speed up model reasoning" (Line 68), which is factually wrong. Speculative decoding applies to all text generation tasks, not just reasoning. In fact, the original speculative decoding (https://arxiv.org/pdf/2211.17192) was evaluated on machine translation and summarization.

  - Line 73-77: this is only for the case where temperature $t>0$. Speculative decoding can also be applied to greedy decoding.

  - The paper claims to propose a "plug-and-play" estimator, when it's clearly not plug-and-play, but requires training. There are actual plug-and-play methods that do the same thing as C2T: https://arxiv.org/abs/2409.00142, https://arxiv.org/abs/2411.18462

- The motivation of this paper is poorly presented, logically flawed, and hardly understandable.

  - The paper states "we conducted entropy-based binning experiments" (Line 106) - what are "entropy-based binning experiments"? Specifically, what's the entropy here? The entropy of the draft model? The entropy of the target model? The cross entropy between them?

  - Line 127-129 use the terms "overestimate" and "underestimate". This implies the paper is trying to compare a "prediction" with a "ground truth value". Yet I have no idea what is being used to predict what, after reading this part at least three times.

  - Line 135: what is $n$? I failed to find its definition anywhere in the paper.

  - **Lemma 2 (Line 142) is not a theoretical contribution from this paper, but directly comes from https://arxiv.org/pdf/2211.17192v2, section 3.2. Yet no credits are given. This is a serious academic misconduct.**

  - Line 176-179: it's obvious that shallower nodes would have higher confidence in draft trees, since the probability for each token must be less than 1, and the confidence of a deeper node is the product of more probabilities. So I have no idea what the paper means by "methods based solely on joint probability cannot distinguish between nodes of different depths using probability alone".

- The method is also not described clearly enough for readers to understand.

  - In Line 202: "labeling each token with its three features" - what exactly are these three features? This is obviously the core of the paper's method, yet it's not properly introduced in the main paper, but only mentioned in the appendix, Line 656.

- The novelty of the proposed method is limited: using a classifier to predict accept probability. This is very similar to many existing works, some of which are even training-free:

  - https://arxiv.org/abs/2405.19715
  - https://arxiv.org/abs/2409.00142
  - https://arxiv.org/abs/2410.18351
  - https://arxiv.org/abs/2411.18462
  - https://arxiv.org/abs/2412.18910

- The empirical experiments do not match the method's theoretical derivation. All derivations are drawn from the scenario where decoding temperature $t>0$, while experiments are conducted with $t=0$ (Line 347).

**Questions:**

It's good to know that the proposed method works on SGLang, though personally I'm more familiar with vLLM. I wonder whether the proposed method works with the continuous batching and other optimizations in vLLM?

**Details Of Ethics Concerns:**

Lemma 2 (Line 142) is not a theoretical contribution from this paper, but directly comes from the well-known speculative decoding paper https://arxiv.org/abs/2211.17192v2, section 3.2. The only difference is that this submission changed the name "natural divergence" from the previous paper to "total variation distance". No credits are given.

---

> ### Author Response · Authors · 2025-11-13
>
> We must seriously disagree with your accusation of serious academic misconduct. First of all, the paper you mentioned is one of the early pioneering works on speculative decoding, so it has been already cited in the introduction section of our work (however, we did not carefully check some of the details inside this paper). Secondly, to clarify, $n$ obviously represents the number of elements in the probability distribution. The mathematical derivation in our Section 3.2 has a clear purpose: **to precisely model the $Bias$ between the expected acceptance rate $E(\alpha)$ and the maximum probability $p_{max}$ in the distribution.**  $TVD$ we defined in the Lemma 2 serves as the bridge connecting these two quantities. **$TVD$ is a commonly-used metric for measuring the distance between two probability distributions**. The simple derivation in Lemma2 is a common sense and straightforward to conduct by the persons with a computer science background.  As for the paper you mentioned, it uses a custom concept called "natural divergence" instead. This difference in terminology led us to inadvertently overlook the similar derivation in these prior works during our paper survey stage, resulting in an unintentional citation oversight at Lemma 2 (We already cited this paper in the introduction section, we can also cite it again here if you think it is necessary). Most importantly, Lemma 2 serves as a necessary prerequisite for deriving **our most important Lemma 3** later, rather than as an independent conclusion and contribution. **Lemma 3 is our contribution**.
>
> We hope you will exercise caution before accusing others of serious academic misconduct!

---

> > ### Comment · Reviewer_surw · 2025-11-13
> >
> > There is no proof or disproof that the authors "inadvertently overlooked the similar derivation" in a paper that's basically the foundation of speculative decoding. I can only state the fact: Lemma 2 is identical to Section 3.2 in https://arxiv.org/pdf/2211.17192v2, yet no credits are given. This is academic misconduct, whether intentional or not.

---

### Official Review · Reviewer_P67C · 2025-10-28

**Soundness:** 3
**Presentation:** 3
**Contribution:** 2
**Rating:** 4
**Confidence:** 5

**Summary:**

Tree-structured approaches like EAGLE-2/3 dynamically expand token trees but can create many candidate tokens, increasing verification cost. The paper proposes reducing the tree size without hurting acceptance length, which pre-prunes candidate tokens during tree construction using a lightweight classifier that scores each node’s confidence based on three statistical features available from the draft model and then use two-stage pruning to integrate with vanilla speculative decoding. The authors verify the effectiveness of proposed method from both theoretical and measured Speedups

**Strengths:**

1. This paper uses draft-model accessible statistics (joint probability, entropy, depth) with a tiny classifier to make effective pruning decisions; low engineering complexity and no change to target model.
2. The proposed method achieves lower candidate token counts while preserving or increasing accept length, directly lowering target-model verification cost and improving throughput.
3. The proposed method generates more compact trees than dynamic methods for comparable accept length, which is valuable under GPU memory/latency limits and reduces operational cost.

**Weaknesses:**

1. The proposed method is highly depended on precise thresholds and top-K, suboptimal tuning can harm recall or lead to larger candidate tokens, reducing speed gains.
2. Pre-pruning can miss tokens ultimately accepted by the target model if classifier recall is insufficient, potentially reducing accept length or increasing verification retries. The paper emphasizes recall in training, but risk remains under distribution shifts.
3. All reported evaluations are at temperature=0; stochastic decoding or different draft/target pairings could change the effectiveness of the classifier features and pruning thresholds.

**Questions:**

How to decide the default threshold and top-K based on the model size?

---

> ### Author Response · Authors · 2025-11-14
>
> Dear Reviewer P67C,
>
> Although we have decided to withdraw our manuscript, I am still very willing to address some of your concerns based on the experimental results we already have. Before the formal withdrawal, we also welcome any further questions you may have.
>
> W1.The proposed method is highly depended on precise thresholds and top-K, suboptimal tuning can harm recall or lead to larger candidate tokens, reducing speed gains.
>
> As shown in Figure 4, the scatter plots for different combinations of topK and $\beta$ almost completely overlap and all outperform the baseline. This is an interesting phenomenon, indicating that under different acceptance lengths, various combinations of $\beta$ and topK achieve similar pruning effects.
>
> W2.Pre-pruning can miss tokens ultimately accepted by the target model if classifier recall is insufficient, potentially reducing accept length or increasing verification retries. The paper emphasizes recall in training, but risk remains under distribution shifts.
>
> The experiments in Appendix D further demonstrate that classifier-based pre-pruning can significantly improve precision compared to EAGLE-2, while maintaining a high recall rate. Of course, it is impossible to achieve 100% recall, but we believe that an average recall rate of 0.9140 is already excellent.
>
> W3.All reported evaluations are at temperature=0; stochastic decoding or different draft/target pairings could change the effectiveness of the classifier features and pruning thresholds.
>
> We will include experiments with temperature=1 in future versions. Thank you for the reminder.
>
> Q1.How to decide the default threshold and top-K based on the model size?
>
> As W1 has responded, as shown in Figure 4, the scatter plots for different combinations of topK and $\beta$ almost completely overlap and all outperform the baseline, indicating that under different acceptance lengths, various combinations of $\beta$ and topK achieve similar pruning effects. Therefore, we can simply fix either $\beta$ or topK and adjust the other parameter to obtain the desired acceptance length. In our practice, we typically fix $\beta$ at 0.5 and adjust topK to achieve different pruning objectives, which should be determined according to model size and resource availability. In our experiments, the default configuration for topK follows EAGLE’s default setting, which is 10.

---

### Official Review · Reviewer_5NQ9 · 2025-10-31

**Soundness:** 3
**Presentation:** 3
**Contribution:** 3
**Rating:** 6
**Confidence:** 5

**Summary:**

This paper presents C2T, which trains a classifier based on joint probability, entropy and depth to construct the candidate token tree for speculative decoding. The author identified 3 problems with the existing joint-prob-based methods and designed the C2T model to mitigate these problems. It is a lightweight, plug-and-play method for improving dynamic tree construction, delivering max 17% speedup over EAGLE2/3. C2T also shows good generality over datasets and models.

**Strengths:**

- This paper is well-written and easy to follow
- The authors conducted in-depth mathematical analysis of the bias problem in the current tree construction methods and provided good explanation for the training process and theoretical performance gain.
- The experiments cover a wide range of model sizes and datasets with consistent improvements.
- Classifier transferability suggests the features (joint probability, entropy, depth) capture model-agnostic properties.

**Weaknesses:**

- The impact of different β/top-K tuning is not clear, some quantitative results to demonstrate their impacts could be provided.

**Questions:**

- why is the number of candidate tokens larger with SD+C compared to SD alone in Table2?
- Table 3 shows speedup even at batch size = 256, can you provide the runtime breakdown, e.g. the wall time of each step in inference?

---

> ### Author Response · Authors · 2025-11-14
>
> Dear Reviewer 5NQ9,
>
> Although we have decided to withdraw our manuscript, I am still very willing to address some of your concerns based on the experimental results we already have. Before the formal withdrawal, we also welcome any further questions you may have.
>
> W1.The impact of different β/top-K tuning is not clear, some quantitative results to demonstrate their impacts could be provided.
>
> As shown in Figure 4, the scatter plots for different topK and $\beta$ combinations almost completely overlap. This is an interesting phenomenon, indicating that under different acceptance lengths, various combinations of $\beta$ and topK achieve similar pruning effects. Therefore, we can simply fix either $\beta$ or topK and adjust the other parameter to obtain the desired acceptance length. In our practice, we typically fix $\beta$ and then adjust topK. Additionally, we observe that relaxing the secondary topK pruning, or even omitting it entirely (i.e., relying solely on the initial confidence-based pruning), allows C2T to reach a higher upper limit in acceptance length (see the blue scatter points in Figure 4).
>
> Q1.why is the number of candidate tokens larger with SD+C compared to SD alone in Table2?
>
> Because this pre-pruning method becomes an early stopping algorithm in the SD (i.e., speculative decoding with chain mode) scenario, if we keep the draft length of C2T consistent with the baseline, the acceptance length of C2T will always be less than or equal to that of the baseline. Therefore, as mentioned in line 345, to better demonstrate the performance of C2T, we increase the draft length by 2 relative to the baseline (from 5 to 7). Of course, this setup may result in potentially unfair comparisons, so here we provide more detailed supplementary experiments (with early stopping upper limit = 6), showing the performance of C2T when the average number of draft tokens per step $\overline \gamma$ is less than that of the baseline. MT, HE, and GS denote MT-Bench, HumanEval, and GSM8K, respectively:
>
> |         |        | MT                 |          | HE                 |          | GS                 |          |
> | :------ | :----- | :----------------- | :------- | :----------------- | :------- | :----------------- | :------- |
> | Model   | Method | $\overline \gamma$ | $\tau$   | $\overline \gamma$ | $\tau$   | $\overline \gamma$ | $\tau$   |
> | L31 8B  | SD     | 5.0                | 3.69     | 5.0                | 4.21     | 5.0                | 3.71     |
> |         |        | 6.0                | 3.81     | 6.0                | 4.64     | 6.0                | 4.05     |
> |         | SD+C   | **5.6**            | **4.06** | **5.4**            | **4.86** | **5.2**            | **4.13** |
> | L33 70B | SD     | 5.0                | 3.72     | 5.0                | 4.26     | 5.0                | 3.82     |
> |         |        | 6.0                | 4.04     | 6.0                | 4.51     | 6.0                | 4.21     |
> |         | SD+C   | **5.2**            | **4.11** | **5.8**            | **4.87** | **5.0**            | **4.26** |
>
> Q2.Table 3 shows speedup even at batch size = 256, can you provide the runtime breakdown, e.g. the wall time of each step in inference?
>
> Thank you for your suggestion. We will consider profiling the wall time distribution during inference in detail in future work.

---

### Note · Authors · 2025-11-27

**Comment:**

We have decided to withdraw our submission to incorporate further improvements. We thank the reviewers for their time.

**Withdrawal Confirmation:**

I have read and agree with the venue's withdrawal policy on behalf of myself and my co-authors.